Meiogyne oligocarpa (Annonaceae), a new species from Yunnan, China

Xue Bine 1
Shao Yun-Yun 2
Xiao Chun-Fen 3
Liu Ming-Fai 4
Li Yongquan yongquanli@zhku.edu.cn 1
Tan Yun-Hong tyh@xtbg.org.cn 5 6
1 College of Horticulture and Landscape Architecture, Zhongkai University of Agriculture and Engineering , Guangzhou , Guangdong , China
2 Guangdong Provincial Key Laboratory of Digital Botanical Garden, South China Botanical Garden , Guangzhou , Guangdong , China
3 Horticulture Department, Xishuangbanna Tropical Botanical Garden, Chinese Academy of Sciences , Menglun, Mengla , Yunnan , China
4 Division of Ecology and Biodiversity, School of Biological Sciences, The University of Hong Kong , Hong Kong , China
5 Southeast Asia Biodiversity Research Institute, Chinese Academy of Sciences & Center for Integrative Conservation, Xishuangbanna Tropical Botanical Garden, Chinese Academy of Sciences , Menglun, Mengla , Yunnan , China
6 Center of Conservation Biology, Core Botanical Gardens, Chinese Academy of Sciences , Menglun, Mengla , Yunnan , China
Sosa Victoria
Electronic publication date: 2021 Apr 14
Publication date: 2021
Volume: 9
Electronic Location ID: e10999
Received 2020 Oct 5; Accepted 2021 Feb 2
Copyright: ©2021 Xue et al.
Copyright year: 2021
Copyright holder: Xue et al.
License: This is an open access article distributed under the terms of the Creative Commons Attribution License, which permits unrestricted use, distribution, reproduction and adaptation in any medium and for any purpose provided that it is properly attributed. For attribution, the original author(s), title, publication source (PeerJ) and either DOI or URL of the article must be cited.
License URL: https://creativecommons.org/licenses/by/4.0/

Keywords: Meiogyne, Molecular phylogeny, Taxonomy, Yunnan, China

Funding: National Natural Science Foundation of China 31872646 31970223 Forestry Scientific Technology Innovation Project of Guangdong Province 2020KJCX010 Foundation of Southeast Asia Biodiversity Research Institute, Chinese Academy of Sciences Y4ZK111B01 Lancang-Mekong Cooperation (LMC) Special Fund (Biodiversity Monitoring and Network Construction CAS 135 program 2017XTBG-F03 This research was supported by the National Natural Science Foundation of China [Grant no. 31872646 awarded to Bine Xue and Grant no. 31970223 awarded to Yunhong Tan], Forestry Scientific Technology Innovation Project of Guangdong Province (No.2020KJCX010), the Foundation of Southeast Asia Biodiversity Research Institute, Chinese Academy of Sciences (Y4ZK111B01), a project of the Lancang-Mekong Cooperation (LMC) Special Fund (Biodiversity Monitoring and Network Construction along Lancang-Mekong River Basin project) and the CAS 135 program (No. 2017XTBG-F03). The funders had no role in study design, data collection and analysis, decision to publish, or preparation of the manuscript.

==============================
Meiogyne oligocarpa sp. nov. (Annonaceae) is described from Yunnan Province in Southwest China. It is easily distinguished from all previously described Meiogyne species by the possession of up to four carpels per flower, its bilobed, sparsely hairy stigma, biseriate ovules and cylindrical monocarps with a beaked apex. A phylogenetic analysis was conducted to confirm the placement of this new species within Meiogyne. Meiogyne oligocarpa represents the second species of Meiogyne in China: a key to the species of Meiogyne in China is provided to distinguish it from Meiogyne hainanensis. Paraffin sectioning was undertaken to study the anatomy of the corrugations on the inner petals of Meiogyne oligocarpa to verify whether they are glandular.

Introduction

The Annonaceae is the most genus-rich and species-rich family in Magnoliales, with ca. 110 genera and ca. 2400 species (Chatrou et al., 2012; Guo et al., 2017; Xue et al., 2020). Meiogyne Miq. is a medium-sized genus in the tribe Miliuseae, subfam. Malmeoideae, currently comprising 32 species of trees or shrubs, distributed in wet tropical lowland and lower montane rainforests across South-east Asia and the western Pacific (Thomas et al., 2012; Xue et al., 2014; Xue, Liu & Saunders, 2017; Turner & Utteridge, 2015; Johnson et al., 2019). It has recently been expanded following the inclusion of species formerly classified in Ancana F. Muell., Ararocarpus Scheff., Guamia Merr., Polyaulax Backer, Fitzalania F. Muell., Chieniodendron Tsiang & Li, Oncodostigma Diels and several specie in Desmos, Uvaria and Polyalthia (van Heusden, 1994, van Heusden, 1996; Turner, 2009; Thomas et al., 2012; Xue et al., 2014; Xue, Liu & Saunders, 2017; Turner & Utteridge, 2015). Among those genera, the name Fitzalania (Mueller, 1863) antedates that of Meiogyne (Miquel, 1865); Chaowasku, Zijlstra & Chatrou (2011) consequently proposed conservation of the latter name (subsequently accepted by the Nomenclature Committee for Vascular Plants: Applequist, 2012).

Yunnan Province is located in southwestern China and harbors more than 19,000 plant taxa, accounting for over 50% of China’s overall floristic diversity (Qian et al., 2020). Plant diversity in Yunnan faces continuous threats with the deterioration of ecology and environment, however. In order to constrain the rapid loss of biodiversity, the Chinese Academy of Science and Yunnan Provincial government jointly launched a project named Tropical Plant Resource Conservation and Sustainable Use from 2000 to 2004. In this project, more than 6,000 plant species collected from tropical areas of China and nearby countries were preserved in 35 living collections in Xishuangbanna Tropical Botanical Garden. These include about 100 Annonaceae collections, among which twelve small trees have been continuously flowering and fruiting in recent years. These treelets were propagated from 20 seeds collected from one unidentified Annonaceae tree in He-kou County, Yunnan Province in 2001 by a field collection team leading by Professor Guo-Da Tao. The flower morphology indicates that it belongs to Meiogyne, but it differs from all previously described species. It is readily distinguished from other Meiogyne species by a combination of the following characters: flowers with bilobed stigmas, up to four carpels per flower, ovaries with ovules attached in two rows and cylindrical monocarps with a beaked apex. Morphological comparisons and phylogenetic analyses based on seven chloroplast regions indicate that the treelets represent a hitherto undescribed species, which we describe and name here as Meigoyne oligocarpa.

Meiogyne is characterized by inner petals with a longitudinally grooved or verrucose base to the adaxial surface and innermost stamens with tongue-shaped apical prolongations (van Heusden, 1994; Thomas et al., 2012; Xue et al., 2014; Johnson et al., 2019). The elaborate inner petal corrugations are synapomorphic for Meiogyne, and have often been referred to as “glands” (van Heusden, 1992; Saunders, 2010), although glandular function has never been confirmed as no liquid secretions have been observed (Xue et al., 2017). Shao & Xu (2015) similarly failed to observe apertures for secretion on the surface of the corrugations (“strumae”) on inner petals of Meiogyne hainanensis (as “Oncodostigma”). They found polysaccharides on the strumae, suggesting that the structure may provide nutrition to floral visitors as well as a protected site for mating, oviposition, brooding and larval feeding (Shao & Xu, 2015). This is consistent with the “food body” hypothesis in Sapranthus (Schatz, 1987) or “nutritious tissues” hypothesis (Gottsberger & Webber, 2018). Moreover, Shao & Xu (2015) also introduced a third hypothesis, the “brood-site” hypothesis. The study failed to establish that the observed floral visitors (thrips) are effective pollinators, however.

In this study, paraffin sectioning was undertaken to investigate the anatomy of the corrugations at the base of the adaxial surface of the inner petals, with the aim to testing the alternative hypotheses regarding the function of the corrugations.

Materials & Methods

Ethics statements

The new species reported in this study was collected from Xishuangbanna Tropical Botanical Garden, Yunnan Province, China, which permitted our field work in the Garden. Since this species is currently undescribed, it is not currently included in the China Species Red List (Wang & Xie, 2004).

Nomenclature

The electronic version of this article in Portable Document Format (PDF) will represent a published work according to the International Code of Nomenclature for algae, fungi, and plants (ICN), and hence the new names contained in the electronic version are effectively published under that Code from the electronic edition alone. In addition, new names contained in this work which have been issued with identifiers by IPNI will eventually be made available to the Global Names Index. The IPNI LSIDs can be resolved and the associated information viewed through any standard web browser by appending the LSID contained in this publication to the prefix “http://ipni.org/”. The online version of this work is archived and available from the following digital repositories: PeerJ, PubMed Central, and CLOCKSS.

Material collection

The new species has been monitored in Xishuangbanna Tropical Botanical Garden by the authors continuously since 2014. Flowering and fruiting specimens were collected for morphological study. Mature flowers for anatomical study were fixed in FAA (70% alcohol, formaldehyde and glacial acetic acid in a ratio of 90: 5: 5) for 24 h and then transferred to store in 70% alcohol. Leaf materials for DNA extraction were collected and dried using silica gel in the field.

Morphological observations

Morphological description of the new species was based on careful examination of materials collected. Comparisons with other similar Meiogyne species were based on the existing literature (van Heusden, 1994; Li & Gilbert, 2011; Thomas et al., 2012; Johnson et al., 2019) as well as the study of herbarium specimens and digitized images (mainly from HITBC, IBSC, KEP, KUN, PE and SING herbaria).

Flower samples in 70% alcohol were prepared for scanning electron microscopy by dehydration and critical-point drying. Carpels, stamens and pollens were then mounted on metal stubs, sputter-coated with gold, and examined using scanning electron microscope (SEM) as in Xue et al., 2017. Flowers fixed in 70% alcohol were also dissected for anatomical observations using paraffin sectioning following Xue et al., 2017.

Molecular phylogenetic analyses

One accession of the new species (Y.Y. Shao SYY26, IBSC), one accession of Meiogyne kanthanensis (J.P.C. Tan et al. FRI81800, KEP), and one accession of M. hainanensis (B. Xue XB293, IBSC) from China were sampled. Seven chloroplast DNA regions (matK, ndhF, ndhF-rpl32, rbcL, rpl32-trnL, trnL-F and ycf1) were sequenced for the three species. These newly generated sequences were added to the seven-region dataset compiled by Xue et al. (2014). The final dataset therefore included 73 Annonaceae accessions, with the ingroup consisting of 30 accessions (representing 26 species) of Meiogyne. For detailed information regarding DNA extraction, PCR amplification, primer sequences and sequence alignment, refer to Thomas et al. (2012) and Xue et al. (2014). The sampled species, voucher information, and GenBank accession numbers are listed in Appendix S1 .

The phylogenetic trees were reconstructed using maximum parsimony (MP) (Swofford, 2003) and Bayesian inference (BI) (Ronquist & Huelsenbeck, 2003) based on the seven combined regions. MrModeltest ver. 2.3 (Nylander, 2004) was used for best-fit likelihood model selection for each region under Akaike Information Criterion: the general time-reversible model with a gamma distribution of substitution rates (GTR+G) was chosen for the matK, trnL-F and ndhF-rpl32 regions; and the GTR+I+G model with a proportion of invariant sites was selected for the ndhF, rbcL, rpl32-trnL and ycf1 regions. For detailed methods in tree reconstruction, refer to Thomas et al. (2012) and Xue et al. (2014).

Results

Phylogenetic analysis

The seven-region concatenated alignment of the 73-accesion dataset consisted of 8,923 characters. The resultant MP and BI topologies based on the concatenated alignment are similar. The BI tree with both posterior probabilities and MP bootstrap values for each clade is shown as Fig. 1. The new species, Meiogyne oligocarpa, is deeply nested within the Meiogyne clade (PP = 1, MPBS = 92%). Although these results confirm that the new species unequivocally belongs to the genus Meiogyne, limitations in internal resolution and support preclude any definitive conclusion regarding which species is phylogenetically closest to M. oligocarpa.

Figure 1 Molecular result showing the phylogenetic position of Meiogyne oligocarpa.

Bayesian 50% majority-rule consensus tree under partitioned models (cpDNA data: matK, ndhF, rbcL, trnL-F, ndhF-rpl32, rpl32-trnL and ycf1; 73 accessions). Numbers at the nodes indicate Bayesian posterior probabilities and maximum parsimony bootstrap values (>50%), in that order.

Figure 2 Flower and fruit morphology of Meiogyne oligocarpa sp. nov.

(A) Lateral view of the flower. (B) Abaxial view of the flower. (C) Adaxial view of the flower, showing corrugated structure at the base of the inner petals, and the bilobed stigma. (D) Adaxial and abaxial surface of the inner petals, showing the corrugated structure at the base of the adaxail surface of the inner petals, and the pubescent indumentum at the abaxial surface. (E) Flower with the petals removed, showing the sepals, stamens and stigmas. (F) Fruit with three monocarps. (G) Fruit with one monocarp. (H) Fruit with the pericarp removed, showing the biseriate seed arrangement. (I) Fresh Seed. (J) Dried seeds, showing the raised raphe; and the longitudinal section of the seed, showing the lamelliform endosperm ruminations (B. Yang XTBG0181, HITBC). (A, C, G by Yun-Yun Shao; B, D–F, J by Yun-Hong Tan; H–I by Chun-Fen Xiao).

Figure 3 Morphology of the surface of carpels, stamens and pollens of Meiogyne oligocarpa sp. nov. (scanning electron micrographs).

(A) Carpel, showing the densely hairy ovary and bilobed sparsely hairy stigma. (B) Dissected carpel, showing the biseriate ovules. (C) Innermost stamen, showing the elongated tongue-shaped apical prolongation. (D) Outer stamen. (E) Pollen grains. (F) Pollen grain, showing the two germination zones. (G) Rugulate pollen exine ornamentation. (A–G by Yun-Yun Shao).

Figure 4 Line drawing of Meiogyne oligocarpa. sp. nov.

(A) Flowering branch. (B) Flower, lateral view. (C) Sepals, ad- and abaxial view. (D) Outer petals, ad- and abaxial view. (E) Inner petals, ad- and abaxial view; adaxial view showing the corrugations at the base. (F) Innermost stamen, ad- and abaxial view; adaxial view showing the elongated tongue-shaped apical prolongation. (G) Outer stamen, ad- and abaxial view. (H) Carpel, showing bilobed stigma. (I) Carpel, longitudinal section, showing the biseriate ovule arrangement. (J) Fresh fruit. (K) Dried fruit, showing monocarp shallowly transversely constricted between seed. (L) Seed. Drawn by Ding-Han Cui, from C. F. Xiao C400733 (IBSC).

Figure 5 Anatomical structure of the carpel, stamen, inner petal and outer petal (paraffin sections).

(A) Section of the carpel, showing the biseriate ovule arrangement. (B) Section of the stamen, showing the four pollen sacs. (C) Section of the corrugated part of the inner petal. (D) Close-up of the section of the corrugated part of the inner petal, showing many cells containing dark-blue stained starch granules and a few number of cells filled with brown-colored tannis mixed with starch granules. (E) Section of the non-corrugated part of the inner petal. (F) Section of the outer petal. (A–F by Yun-Yun Shao).

Morphological comparisons

Meiogyne oligocarpa has basally and adaxially grooved inner petals (Figs. 2C, 2D) and the innermost whorl of stamens has expanded connectives (Fig. 3C). These characters are congruent with its placement in the genus Meiogyne. Meiogyne oligocarpa is distinct, however, in having a combination of the following characters: bilobed stigmas (Figs. 2C, 2E, 3A and 4H), up to four carpels per flower (Figs. 2C, 2E), cylindrical monocarps with a beaked apex (Figs. 2F, 2G, 3J and 3K), and seeds in two series (Figs. 2H, 3B and 5A).

Meiogyne oligocarpa is most similar to M. kanthanensis Ummul-Nazrah & J.P.C.Tan from Perak, Malaysia and Southern Thailand in its overall flower morphology: both species have large, broad and pubescent petals. The two species differ, however, as follows. Meiogyne oligocarpa has smaller leaves (12–17 × 3–4.2 cm) with 10–13 veins on each side of the leaf, compared with M. kanthanensis, which has larger leaves (16.4–22. 6 × 6.1–9 cm) with 8–10 pairs of veins (Tan et al., 2014; Johnson et al., 2019). Meiogyne oligocarpa has 1–4 carpels per flower with bilobed, sparsely hairy stigmas, whereas M. kanthanensis has 3–5 carpels with globose, densely hairy stigmas (Tan et al., 2014; Johnson et al., 2019). The monocarps of M. oligocarpa are cylindrical (6–8 ×1.5–2 cm) with a beaked apex, whereas those of M. kanthanensis are oblong (2.7–5.5 × 2.3–3.2 cm) with a rounded apex (Johnson et al., 2019). A detailed morphological comparison of the two species is summarized as Table 1. Our phylogeny nevertheless indicates that the two species are not closely related (Fig. 1).

To conclude, both the molecular and morphological data support the placement of the new species in Meiogyne. It differs from all previously described species, and therefore unequivocally represents a new species.

Taxonomic treatment

Meiogyne oligocarpa B.Xue & Y.H.Tan, sp. nov. (Figs. 2–5)

Type. CHINA. Yunnan Province, He-kou County, the voucher from cultivated plants in Xishuangbanna Tropical Botanical Garden, 21°55′25″N, 101°16′13″E, alt. 586 m. C. F. Xiao C400733, 15 Oct. 2016 (fl., fr.) (holotype HITBC; isotype IBSC)

Diagnosis. Meiogyne oligocarpa is distinct among Meiogyne species in having a combination of characters, including up to four carpels per flower, bilobed sparsely hairy stigmas, cylindrical monocarps with a beaked apex, and seeds in two series. It is most similar to Meiogyne kanthanensis, but differs in having smaller leaves with more secondary veins, shorter pedicels, stigmas that are bilobed and distinctly less hairy, and cylindrical monocarps with a beaked apex.

Description. Small trees to 5 m tall, ca. 4 cm dbh. Bark grayish. Young twigs green, yellowish puberulent, soon become grayish and glabrous. Petioles 3–5 mm long, 1–2 mm in diameter, pubescent; leaf laminas lanceolate, or narrowly elliptic or narrowly oblong, length:width ratio 3.5–4.5, 12–20 × 3–4.2 cm, base slightly asymmetrical, oblique to rounded, apex acute to acuminate, papery, slightly glossy above in vivo, drying dull greyish-green, concolorous beneath, glabrous above, beneath base of leaf margin and midrib pilose, secondary veins sparsely pilose; midrib impressed and glabrous above, raised and hairy below; secondary veins 8–13 on each side of the leaf, parallel, diverging at 45–60°from midrib, upturned and gradually diminishing towards apex, connecting to subsequent secondary veins by series of cross veins, and lacking prominent marginal loops, distinctly raised below; tertiary veins scalariform, prominent abaxially (Fig. 4A). Inflorescences axillary; one flower per inflorescence (Figs. 2B, 4A), greenish to yellowish with a strong fruity scent in vivo. Peduncles absent (Figs. 2B, 4A). Pedicels 5–8 mm long, 1–1.5 mm in diameter, pubescent, with two minute pubescent bracts (Figs. 2B and 4B). Sepals triangular, 7–8 × 4–5 mm, pubescent abaxially, glabrous adaxially (Figs. 2B, 2E and 4C); outer petals ovate to lanceolate, 42–50 × 15–17 mm in vivo, 30–45 × 9–15 mm when dry, apex acute, grayish pubescent abaxially, glabrous to sparsely villous adaxially (Figs. 2A–2C, 4D); inner petals narrower, shorter, cohering when young, 37–48 × 14–17 mm in vivo, 26–38 × 9–12 mm when dry, grayish pubescent abaxially, glabrous to sparsely villous adaxially, basally grooved, often turning reddish when mature, fragrant, margin slightly recurved (Figs. 2A–2D, 4E). Stamens with shield-like connective prolongation, elongated in inner whorl, 50–56 per flower, ca. two mm long (Figs. 3C, 3D, 4F, 4G, 5B). Carpels 1–4 per flower, ca. four mm long, with long gray hairs (Figs. 3A, 4H); stigmas shallowly to deeply bilobed, lobes less obvious when dried, sparsely hairy (Figs. 2C, 2E, 3A, 4H); ovules 12–20 per carpel, biseriate (Figs. 3B, 4I, 5A). Fruiting pedicels 3–5 mm long, three mm in diameter; monocarps 1–3 per fruit, smooth, cylindrical, ca. 6–8 cm long, 1.5–2 cm in diameter, apex rostrate, beak 5–10 mm, base contracted into stipe 3–10 mm, densely blackish pubescent, shallowly transversely constricted between seeds when dried (Figs. 2F, 2G, 4J and 4K). Seeds reniform to ellipsoid, up to 20 per monocarp, biseriate, 1.5 cm long, 1 cm wide, 8 mm thick, surface smooth in vivo, wrinkled when dried, raphe raised, endosperm ruminations lamelliform (Figs. 2H–2J, 4L). Pollen grains solitary, subspherical, disulculate, 30–40 µm in diameter, rugulate (Figs. 3E–3G).

Etymology. The specific epithet reflects that the flower has few carpels (1–4) and hence the fruit has few monocarps (1–3).

Paratypes. China. Yunnan: Ge-jiu City, Man-hao, alt. 700m, 28 Apr. 1994, Ge-jiu Forestry Bureau exped. 94111 (KUN); Meng-la County, Men-lun, cultivated in Xishuangbanna Tropical Botanical Garden, 6 Mar. 2006, W.Q. Xiao C400431 (HITBC); 25 Nov. 2009, W.Q. Xiao C400583 (HITBC); 23 Mar. 2015, Y.Y. Shao SYY26, SYY27 (IBSC); 13 Oct. 2015, Y.Y. Shao, SYY28 (IBSC, HKU); 5 Aug. 2017, Y.H. Tan & H.B. Ding XTBG0007 (HITBC); 3 May 2020, B. Yang XTBG0104 (HITBC); 26 Aug. 2020, B. Yang XTBG0181 (HITBC).

Distribution. Only known from 12 individuals cultivated in Xishuangbanna Tropical Botanical Gardens, Yunnan, China. According to records, it was introduced from He-kou County, Yunnan Province in 2001. In our extensive study of Annonaceae herbarium collections in China, we find only one specimen from Man-hao, Ge-jiu city (Ge-jiu Forestry Bureau exped. 94111, KUN) representing this species collected from wild populations. One of the authors, Yun-Hong Tan, has undertaken extensive field survey in He-kou County and adjacent regions, but has failed to locate any wild populations. Primary forests in He-kou have been under severe pressure from agricultural expansion over recent decades, and most unprotected forests at low elevation have been replaced by banana plantations. Additional field surveys are required to locate wild populations of this species in Yunnan. This study highlights the essential role of botanical gardens on biodiversity conservation.

Ecology and phenology. In evergreen forests. Flowering and fruiting from March to September.

Table 1 Morphological comparison among Meiogyne oliocarpa, M. kanthaensis and M. hainanensis.

Characters	Meiogyne oliocarpa	M. kanthaensis	M. hainanensis	
Size of the leaves	12–17 × 3–4.2 cm	16.4–22.6 × 6.1–9 cm	4–16 × 1.5–5 cm	
Number of secondary veins on leaf	10–13 pairs	8–10 pairs	6–9 pairs	
Size of outer petals when dry	30–45 × 9–15 mm	42 × 13–23 mm	9–13 × 5–8 mm	
Size of inner petals when dry	26–38 × 9–12 mm	37–41 × 15–17 mm	8–12 × 5–7 mm	
Number of carpels per flower	1–4	3–5	up to 10	
Shape of the stigmas	bilobed	globose	clavate to conical	
Indumentum of the stigmas	sparsely hairy	densely hairy	glabrous	
Size and shape of the monocarps	cylindrical, with a beaked apex, 6–8 ×1.5–2 cm	unknown	cylindrical, ovoid or ellipsoid, apex often rounded, 0.7–7 ×0.7–4.8 cm	
Indumentum of the monocarps	blackish pubescent	unknown	rusty pubescent	

Discussion

Anatomy of inner petal corrugations

The elaborate inner petal corrugations are synapomorphic for Meiogyne. Although these structures have previously often been referred to as “glands” (van Heusden, 1992; Saunders, 2010), glandular function has never been confirmed (Xue et al., 2017; Saunders, 2020). Shao & Xu (2015) examined the inner petal of Meiogyne hainanensis (as “Oncodostigma”) but found no evidence of secretion. It is therefore doubtful whether the corrugations represent true glands, although additional species should be studied.

During our continuous field observations on M. oligocarpa, no liquid secretions were observed to form on the inner petals, although strong fruity scents were emitted during anthesis. The anatomical results also indicate that the corrugations are not glandular. The anatomical organization of nectar glands often consist of four distinct tissues (Nepi, 2007; Xue et al., 2017): epidermis; sub-epidermal secretory parenchyma, comprising several layers of small cells with densely staining cytoplasm; ground parenchyma, comprising several layers of larger cells, more loosely packed than those of the secretory parenchyma; and vascular bundles. The anatomy of the inner petal corrugations of M. oligocarpa (Figs. 5C, 5D), however, consist of an epidermis, several layers of homogeneous parenchyma and a few vascular bundles, and do not differ from adjacent non-corrugated parts of the inner (Fig. 5E) and outer petals (Fig. 5F). The inner petal corrugations are therefore not glandular.

Shao & Xu (2015) studied the corrugations of Meiogyne hainanensis. Based on their observations, they raised two alternative hypotheses regarding the function of the inner petal corrugations, noting that polysaccharides may provide nutrition for floral visitors, and hence the corrugations may function as a “food body” as in Sapranthus (Schatz, 1987), aligned with the “nutritious tissues” hypothesis (Gottsberger & Webber, 2018).

In our study, we observed numerous curculionid beetles within the flowers of M. oligocarpa, with evidence of gnawing of the petals, although this was present on both the corrugated and non-corrugated parts of the inner petals. We also observed starch and tannin in both whorls of petals (Figs. 5C–5F), which may provide food for beetles (Gottsberger & Webber, 2018). A more detailed histochemical study is nevertheless required, as well as additional field observations on the distribution of gnawing marks to verify the “nutritious tissues” hypothesis.

The second hypothesis raised by Shao & Xu (2015) is that strumae may provide floral visitors with a protected site for mating, oviposition, brooding and larval feeding, thus functioning as a “brood-site”. Although this might represent an example of pollinator brood-site adaptations of petals in Annonaceae, Shao & Xu (2015) failed to establish that thrips are the effective pollinator (Saunders, 2020). This also requires further study. It would also be interesting to check whether beetle larvae are present on the petals of M. oligocarpa.

Meiogyne species in China

According to the Flora of China (Li & Gilbert, 2011), only one Meiogyne species is recorded from China: M. kwangtungensis Li, known from only two fruiting collections from Hainan Province. Rainer & Chatrou (2006) suggest that M. kwangtungensis might be better accommodated within Pseuduvaria or Mitrephora, although its placement in Meiogyne is debatable due to the lack of flowers. Fortunately, with new flowering and fruiting collections recently collected from Hainan Province, we have been able to confirm that M. kwangtungensis should be placed in Pseuduvaria (Wang et al., 2021)

The only currently accepted Meiogyne species in China is M. hainanensis (Merr.) Bân from Hainan Province (Tsiang & Li, 1979; van Heusden, 1994; Rainer & Chatrou, 2006; Xue et al., 2014), although Li & Gilbert (2011) treated this species as Chieniodendron hainanense (Merr.) Tsiang & P.T. Li . The transfer of Chieniodendron to Meiogyne has been widely accepted ( Bân 1973; van Heusden, 1994) and supported in recent molecular phylogenetic studies (Thomas et al., 2012; Xue et al., 2014).

Figure 6 Morphological comparison of the two Meiogyne species in China.

(A–C) Meiogyne oligocarpa. (A) Leaf, adaxial view. (B) Inflorescence comprising a solitary flower. (C) Fruit, showing one monocarp a beaked apex. (D–J) Meiogyne hainanensis. (D) Leaf, adaxial view. (E) Inflorescence comprising one to several flowers. (F) Flower, top view. (G) Flower with one outer petal removed, showing the globose clavate stigma. (H) Fruit, showing up to eight monocarps with a rounded apex. (I) Monocarp with pericarp removed, showing biseriate seed arrangement. (J) Seed. (A, by Chun-Fen Xiao; B, C, G by Yun-Yun Shao; D by Bine Xue; E, F, H, by Yun-Hong Tan; I–J by Ren-Bin Zhu).

Meiogyne oligocarpa and M. hainanensis share the possession of biseriate ovules and reniform seeds (Figs. 2H, 2I, 6I 6J). The two Chinese Meiogyne species differ, however, in the number of secondary veins per leaf, the number of flowers per inflorescence, the size of the flowers, the number of carpels per flower, the shape of the stigma, and the indument and shape of the monocarp apex (Table 1). The two species could be easily distinguished from each other vegetatively, since M. oligocarpa has more secondary veins than M. hainanensis (10–13 vs 6–9 pairs per leaf) (Figs. 6A, 6D). When flowers and fruits are available, the differences are even more distinct. Meiogyne oligocarpa has inflorescences comprising a large, solitary flower, 1–4 carpels per flower with bilobed stigmas and fruit with 1–3 beaked monocarps with blackish pubescent indumentum (Figs. 2–4, 6B, 6C). Meiogyne hainanensis has inflorescences with 1–4 relatively smaller, thick and fleshy to leathery petals with a rusty velutinous indumentum (Fig. 6E, 6F), carpels with clavate stigmas (Fig. 6G) and up to eight monocarps with a rusty pubescent indumentum and a rounded apex (Fig. 6H) (Tsiang & Li, 1979; van Heusden, 1994; Li & Gilbert, 2011).

Key to Meiogyne species in China

1a. Leaf laminas with 8–13 pairs of secondary veins; inflorescences comprising a solitary flower; outer petals 30–40 × 9–14 mm when dry, inner petals 26–38 × 9–12 mm when dry, indumentum grayish pubescent; carpels 1–4; stigma bilobed; monocarps 1–3, blackish pubescent, apex with long beak to 10 mm; distributed in He-kou and Ge-jiu, Yunnan. .………………………………………………M. oligocarpa

1b. Leaf laminas with 6–9 pairs of secondary veins; inflorescences comprising 1–4(–8) flowers; outer petals 9–13 × 5–8 mm when dry, inner petals 8–12 × 5–7 mm when dry, indumentum densely rusty velutinous; carpels up to 10, stigma clavate to conical; monocarps to 8, rusty pubescent, apex often rounded; distributed in Hainan. ………………………………………………………………M. hainanensis

Supplemental Information

Supplemental Information 1 List of the 20 new Meiogyne sequences included in this study for phylogenetic analysis, with GenBank accession numbers (matK, ndhF, ndhF-rpl32, rbcL, rpl32-trnL, trnL-F and ycf1)

Click here for additional data file.

Supplemental Information 2 Voucher information and GenBank accession numbers for the 73 Annonaceae samples used in this study

The data are given using the following format: taxon name, origin, collector(s) and collector number (herbarium acronym), matK, ndhF, ndhF-rpl32, rbcL, rpl32-trnL, trnL-F, ycf1 GenBank numbers. Missing data are noted with an —; newly generated sequences for this study are annotated with an*.

Click here for additional data file.

We are grateful to the curators of IBSC, KEP, KUN, HITBC, PE and SING for permission to access their collections. Thanks Dr. Richard Chung and Dr. Saw Leng Guan in KEP for sending type specimen images and leaf material of Meiogyne kanthanensis; Qin Ban in PE for sending specimen photo; Ding-Han Cui for the line drawing; Xiao-Ying Hu for technical assistance with SEM at South China Botanical Garden; Richard Saunders for improving the language of this manuscript; Thomas Couvreur and two anonymous reviewers for the constructive comments to improve this manuscript.

Additional Information and Declarations

Competing Interests

Author Contributions

Data Availability

New Species Registration

The authors declare there are no competing interests.

Bine Xue conceived and designed the experiments, performed the experiments, analyzed the data, prepared figures and/or tables, authored or reviewed drafts of the paper, and approved the final draft.

Yun-Yun Shao and Chun-Fen Xiao performed the experiments, prepared figures and/or tables, and approved the final draft.

Ming-Fai Liu performed the experiments, analyzed the data, authored or reviewed drafts of the paper, and approved the final draft.

Yongquan Li conceived and designed the experiments, authored or reviewed drafts of the paper, and approved the final draft.

Yun-Hong Tan conceived and designed the experiments, performed the experiments, authored or reviewed drafts of the paper, and approved the final draft.

The following information was supplied regarding data availability:

The newly generated sequences are available at GenBank: MW024837–MW024856.

The 20 newly generated sequences, voucher information, and GenBank accession numbers for the 73 Annonaceae samples used in this study are available in Supplemental Files.

The following information was supplied regarding the registration of a newly described species:

Meiogyne oligocarpa B. Xue & Y. H. Tan: 77215102-1

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
