# Peer review of "Meiogyne oligocarpa (Annonaceae), a new species from Yunnan, China"

_PeerJ, doi:10.7717/peerj.10999_

## Round 0.1 · original submission · Minor Revisions

I completely agree with comments by the three reviewers, in which changes in Introduction considering previous research on the family and genus should be added and in Discussion more explanations are needed describing some of the characters. Please take into account the suggestions by the third reviewer.

·

Basic reporting

This article is well written and the English is of good scientific quality. The literature and references are good. I would however add an introduction to the Annonaceae family with some references as the start of the introduction.
The results are relevant and well presented. the hypothesis of a new species is very well supported by different types of data (morphological, anatomical, and molecular to a certain extent).

Experimental design

The article is well designed with a solid experimental design. The authors use different methods and approaches to reach their conclusions.
The methods are well described.
I would have liked to have some more details on where the rest of the genetic data comes from (the markers of the rest of the species, and outgroups).

Validity of the findings

Overall this MS provides sufficient data to support the description of the new species. All data point towards this conclusion. The phylogenetic methods are well explained and robust and sufficient for the conclusions.
The conclusions and discussion are well explained. the key to the two species of Meiogyne is very useful.

Additional comments

This article presents the description of a new species of the genus Meiogyne (Annonaceae). I am not a specialist of the genus or the region, but the data provided are convincing.
This species was mainly described for a few collections growing in the Xishuangbanna Tropical Botanical Garden, which is quite interesting, and unusual for Annonaceae. The authors provide substantial evidence for the status of new species, from morphology, to anatomy and molecular phylogenetics.
I have no major comments, and only minor ones. The methods are appropriate and well done. The English is very good, the figures really nice. The short key to the two species of Meiogyne in China is parallel and precise. Overall a very good and complete manuscript. Congratulations to the authors.
Minor:
Line 35: maybe a short intro to the Annonaceae family?
Line 117: where did the rest of the sampling come from? Could the authors cite a publication? Idem for the outgroups?
Line 209: when and where was this species transferred? Is this transfer official? Please provide some details.
Line 300: “of” secondary veins
Line 305: “inflorescences”, with an s.
308: flowers per se cannot be leathery, do you mean the petals?

Reviewer 2 ·

Basic reporting

'no comment'

Experimental design

'no comment'

Validity of the findings

'no comment'

Additional comments

The manuscript expands the number of Chinese species of Meiogyne from one to two. It's a straightforward thing to do, but the authors have been conscientious in their approach: the phylogenetic placement of the new species within the genus Meiogyne is convincingly demonstrated, the species is accurately described and nicely illustrated.

However, there are some things that I would like to see changed or added before this manuscript would be suitable for publication:

1) Introduction: the introduction should be improved and it is necessary to present hypotheses about the phylogenetic position of the new species, about the anatomy of its petals and about its biogeographic origin.

a) one of the authors' objectives is to infer the function of the longitudinally grooved or verrucose base surface at the inner petals, however, they do not mention anything relevant in the text, beyond it is a characteristic present in all species of the genus Meiogyne. They cited some references but did not mention anything about it. These modifications in the petals are also present in other genera of Miliuseae. Specifically in the genus Sapranthus, a lineage relatively close to Meiogyne (An ancestral character?). Schatz in his doctoral thesis (1987) suggests that those "topographical" modifications in the petals are associated with sites where flower visitors bite (eat? = reward?).
b) The second species described provides the necessary material to hypothesize about the origin and evolution of the genus Meiogyne in China. For example, it could be expected that both species are sisters and that they originated in the same geographical region where their evolution is related to the same factors; or that according to their phylogenetic position the invasion of China by the ancestral lineage happened during a certain period of time and from a certain geographical point. I'd like to see something similar (or other hypotheses) postulated in the introduction.
The results of this manuscript suggest that they are independent invasions and in different periods of time, something very interesting, right?
A previous study (where the first author participated) could be very useful to address the above: see Thomas et al. 2012 where they analyzed the biogeographic history of the genus Meiogyne but including fewer species than those presented here.

2) Materials and methods
a) It is necessary to include dating analysis and reconstruction of ancestral areas (related to the previous comment).
b) Phylogenetic trees (MP, ML, Bayesian) must be presented separately, even if not in the main text.
c) It would be very interesting for the authors to construct a dendrogram of morphological similarity with Meiogyne species (although this is not mandatory from my point of view). If the topology of the dendrogram is similar to that of the phylogenetic tree, this data could be used to include it in the phylogenetic reconstruction and assess if it improves the resolution. What do you think about it?


Results & Discussion
a) Results and discussion must be separated. There are some notes after the taxonomic treatment section (eg "Meiogyne species in China") that are part of the discussion and therefore the paragraphs should be rearranged.
b) The comparison of the new species with its relatives (this could also be derived from the dendrogram) should be in a table for easier tracking. Meiogyne hainanensis should be included in this table.
c) There are three accessions of M. cylindrocarpa with a different phylogenetic position, why not present them as new species?

Reviewer 3 ·

Basic reporting

no comment

Experimental design

no comment

Validity of the findings

no comment

Additional comments

This is a standard manuscript reporting a new species, with detailed description and phylogenetic corroboration. The new species status is warranted by the integrative evidence provided. The anatomical finding is valuable for the understanding of petal evolution in Annonaceae.

Nevertheless, there is a room for the improvement of the manuscript. My comments are embedded in the PDF file of the manuscript, most of them minors. I would like to see a further discussion on the corrugated structures. In the PDF file I have explained this in details.

Annotated reviews are not available for download in order to protect the identity of reviewers who chose to remain anonymous.

---

## Round 0.2 · Minor Revisions

Thank you for considering issues raised previously by the three reviewers. My only concern is that I found a number of misspellings and some problems with English. For example, carpel is misspelled in one of the figure legends, or even names of authors cited in the text are not written correctly (e.g. Couvrier). Some phrases lack verbs (e.g. in the Abstract). Therefore, please use an editorial service or ask a fluent English speaker to review your paper and explicitly indicate this in Acknowledgments.

---

## Round 0.3 · accepted · Accept

The paper is accepted for publication in PeerJ, however during the editorial process indeed please review the paper carefully. I found that carpel is not well written in Table 5 and Table 1 should say among instead of between, among the main points I found.